# Psychological Resilience and Future Anxiety Among University Students: The Mediating Role of Subjective Well-Being

**DOI:** 10.3390/bs15030244

**Published:** 2025-02-20

**Authors:** Alper Bahadır Dalmış, Emrah Büyükatak, Lütfi Sürücü

**Affiliations:** 1Department of Management and Organization, Aeronautical Vocational School of Higher Education, University of Turkish Aeronautical Association, Ankara 06790, Türkiye; abdalmis@thk.edu.tr; 2Department of Education Sciences, Hacettepe University, Ankara 06800, Türkiye; 3Department of Business Administration, World Peace University, Nicosia 99010, Türkiye; lutfi.surucu@wpu.edu.tr

**Keywords:** psychological resilience, future anxiety, subjective well-being, university students

## Abstract

Future anxiety is the worry and concern individuals experience regarding uncertainties and potential negative outcomes in their future. This emotional state can manifest at different stages of students’ academic lives and can impact their academic performance and social relationships. In the process of coping with negative experiences and overcoming challenges, psychological resilience plays a crucial role. Students who struggle to manage stress and have high levels of anxiety tend to experience future anxiety more intensely. The aim of this study is to determine the mediating role of subjective well-being in the relationship between psychological resilience and future anxiety among university students. The study was conducted with a total of 483 university students, including 280 females and 203 males. Data were collected using the Connor–Davidson Resilience Scale (CD-RISC-10), Subjective Well-Being Scale (SWB-7), and Future Anxiety (Dark Future) Scale (Short Form). Analyses were performed using AMOS 22.0 and SPSS 27.0 software. The findings indicate that psychological resilience has a significant negative effect on future anxiety, a significant positive effect on subjective well-being, and that subjective well-being has a significant negative effect on future anxiety. Additionally, the study found that subjective well-being mediates the relationship between psychological resilience and future anxiety.

## 1. Introduction

One of the fundamental elements for the future of a country is its students. They are the guarantee of the future and make the most important contribution to the future for configuration and development of the countries, regions, and groups they belong to. It is evident that countries which have a good education system and offer good education opportunities to their students are developing, while those that cannot provide these opportunities are lagging behind. In parallel with this, the contribution of students to the economic power of their countries is quite valuable. It is stated that higher education plays an increasingly critical role in the economic competitiveness of local and national economies, and higher education, one of the factors of the World Economic Forum’s Global Competitiveness Index, is starting to be considered as a driver of economic growth ([63]). The benefits that students will provide, especially to developing countries, are too much to deny. Students and educational institutions, especially universities, increase the welfare level and quality of life of society.

Recently, every sector and every individual are affected by the increasing problems and traumatic situations caused by wars, violent events, natural disasters, terrorist attacks, and economic crises. Education sector and students, who are the main element of education, are also affected. As a natural result of all these negative developments and rapid technological advances, students’ uncertainties about their future increase due to the changes occurring in countries. When students’ future plans, job, education expectations, and the status of economic opportunities are taken into account as a whole, anxiety, especially future anxiety occurs due to the increasing uncertainty and risk.

Anxiety is expressed as one of the phenomena that characterizes the 21st century due to the stresses that a person encounters in every aspect of life. Future anxiety, first expressed by [103] ([103]), is a pessimistic view of the future in which negative thoughts overshadow positive thoughts, leading to overwhelming negativity. The individual, by nature, has many constant expectations that create a feeling of anxiety about future events. Therefore, all kinds of anxiety have a relatively short-term effect, but future anxiety appears to be ongoing and long-term ([103]).

Students who have anxiety about the future have a state of concern, fear, and anxiety that negative changes will occur in the future, and they have a more negative perspective on life because they think that they cannot control their future because of their negative experiences and there are potential threats. If students have difficulty overcoming the stress they experience and high anxiety levels, they feel anxiety about the future more. However, students need to cope with the negative situations they experience in every period of their education, adapt to the conditions they encounter, and be hopeful about the future. Psychological resilience becomes important in the process of overcoming these difficulties they encountered and adapting to this situation. The fact that students make an effort to cope with negativities and stand strong shows that they are psychologically resilient.

Although various definitions have been proposed ([55]), psychological resilience is the mechanism of coping with and overcoming difficult conditions, a person’s successful adjustment to change, resisting the negative effects of stress factors, and avoiding major dysfunctions ([87]; [101]). It is the process of being able to respond well to these challenges mentally, spiritually, emotionally, physically, financially, or socially, and adjustment to changing conditions calmly and competently by taking advantage of existing strengths ([84]). Psychological resilience is the ability to recover quickly from major changes, depression, severe illnesses, expectations and fears about the future, or negative experiences. It is the ability to easily return to the previous state after negative experiences as well ([71]). Thus, it indicates the process of individuals successfully coping with the psychological state that arises as a result of adverse conditions and returning to their old life.

Individuals who have high resilience are optimistic about the world, and difficult and stressful situations are often treated as a challenge and a new experience for them ([64]). It is thought that students with high psychological resilience can successfully cope with trauma, adverse living conditions or risky conditions, and they adapt to new conditions easily. [62] ([62]) states that individuals with high psychological resilience have characteristics such as problem solving, social competence, autonomy, and positive future expectations. In addition, students with high hope levels and expectations about the future are more likely to endure stress or problems ([75]). Thus, it is seen that individuals’ psychological resilience has an effect on future anxiety.

Along with the effects of psychological resilience on individuals’ future anxiety, it is important to examine other concepts that predict psychological resilience and may also affect future anxiety. One of these concepts is subjective well-being. Subjective well-being is a person’s subjective feelings such as contentment, happiness, satisfaction with life, work, success, usefulness, sense of belonging, the absence of distress, dissatisfaction, or anxiety, etc. Subjective well-being can be defined as an individual’s quality of life in terms of the presence and relative frequency of both positive and negative emotions over time, as well as the person’s general satisfaction with life ([23]).

It has been found that individuals with high levels of subjective well-being are more social and creative ([23]), earn more money, are more productive in business life ([67]), and cope better with stress ([26]). These explanations also show that the subjective well-being of students will affect the future welfare level of a country. By identifying factors that affect subjective well-being and reducing these factors that negatively affect them, students can be happier in the future, and thus they will not have anxiety about the future, because students have a tendency to look to the future with confidence when their subjective well-being increases.

In general, psychological resilience of students includes showing flexibility in the face of negative events, staying strong, and adapting to new situations. However, achieving and maintaining this situation is affected by many factors. These factors are external factors that express environmental and social effects such as wars, natural disasters, and economic crises ([40]; [73]) as mentioned above. The environment of uncertainty created by these environmental factors causes students’ psychological resilience and subjective well-being to be low. Students’ concerns about their future increase due to the effect of these negative developments and changes. Therefore, it is important to have a stable and safe environment for students. The fact that students have a stable and safe environment during school years contributes to the development of their resilience and, consequently, their subjective well-being. This situation is associated with a high level of psychological resilience and psychological well-being ([21]). Based on this information and findings, it can be stated that future anxiety and subjective well-being are concepts that may have a significant relationship with psychological resilience, and subjective well-being has a significant mediating role in the relationship between psychological resilience and future anxiety.

The main purpose of the study is to determine the mediating role of subjective well-being in the relationship between psychological resilience and future anxiety of university students. Our study makes many contributions to practitioners and academic literature. First of all, although there are studies on psychological resilience, subjective well-being, and future anxiety in the literature ([23]; [103]; [40]; [95]), there are mostly studies examining the effect of psychological resilience on subjective well-being or the relationships between psychological resilience and future anxiety ([70]; [35]; [99]; [82]). When viewed from this aspect, our study will contribute to the literature and bring innovation due to both the limited number of studies in this field in the literature and the limited number of studies conducted on university students. Secondly, our study will provide conceptual and theoretical contributions to a better understanding of the concepts of psychological resilience, future anxiety, and subjective well-being. Although it is known that these concepts are related to each other, our study will fully determine whether there is a mediating effect on the relationship levels of these variables. Thirdly, it is expected that the determination of the relationships between the concepts of psychological resilience, future anxiety, and subjective well-being will affect future studies. In order to have a good future, we can plan the future by understanding our own situation and the changes that occur. We can easily adapt to changes by predicting the conditions that will occur in the future. Therefore, future studies are very important. With this respect, our future-oriented and future-oriented study will make many contributions to practitioners and academic literature.

## 2. Theoretical Framework

### 2.1. Psychological Resilience

Although research on psychological resilience has gained momentum in the last two decades, there is still no generally accepted definition of psychological resilience. In general, psychological resilience refers to the ability of adults exposed to a potentially devastating event to maintain stable (such as the death of a loved one, exposure to a violent event, etc.), and capability of healthy levels of psychological and physical functioning, as well as the capacity for positive emotions ([11]).

There are many changes in the lives of university students during their education, such as living away from their families, changing friends and cities, and experiencing financial difficulties. Overcoming these situations is a very difficult process, especially for university students, but students with high levels of resilience achieve more positive results. It is known that students who adapt to extraordinary conditions and achieve positive and unexpected results in difficult situations have high psychological resilience ([42]). In this context, how students cope with the chaotic situations and how they get rid of the difficulties they experience by adjusting is explained with psychological resilience.

### 2.2. Subjective Well-Being

In the literature, subjective well-being is defined as emotional and cognitive evaluations that include feelings such as happiness, peace, satisfaction, and life satisfaction regarding an individual’s life ([27]). In other words, it is a general assessment of the feelings and thoughts that an individual has about his life ([93]; [46]).

For students, subjective well-being is the result of their own assessments of situations in normal life fluency and their reactions in this direction. When the factors affecting the subjective well-being of university students are evaluated, purposeful life activities such as determining their life goals and achieving them come to the fore. Having a good job and the desire to have an easy and comfortable life where daily needs are met are some of the most important purposeful life activities of students. Meeting their basic needs and having a comfortable and good life affect subjective well-being and ensure the emergence of positive emotions. So, it is expected that students with high subjective well-being tend to report more positive emotions and satisfaction with life and less negative emotions ([25]).

### 2.3. Future Anxiety

Future anxiety is a special type of anxiety and is a state of inability to endure uncertainties that will occur due to changes involving risks for the personal future, and it is feeling fear and anxiety ([59]). Future anxiety also is a form of conscious anxiety that can affect thoughts, feelings, and behaviors. It is way of increasing fear of future events and anticipation of changes perceived as dangerous or negative. People with high future anxiety often fear global catastrophes (e.g., earthquakes or war) or painful personal experiences ([91]).

A feeling of high levels of anxiety about a future event or situation that has not yet occurred is a tendency to focus heavily on negative thoughts and outcomes. Future anxiety causes individuals to avoid making plans for the future. In today’s societies, where competition and success are at the forefront, future anxiety begins at a very early age for students.

Students experience anxiety, especially during their university years, since their future will be shaped. During the transition from adolescence to adulthood, young adults have goals of creating a professional identity and establishing close relationships ([80]). The inability to achieve independence and these developmental goals arouses stress and anxiety in this age group ([6]; [50]). [9] ([9]) stated that university students are more anxious about their future than people at other developmental stages of life. These concerns most likely arise when high expectations regarding education and relationships do not match real life, and this creates negative thinking, reduces life satisfaction, and ultimately leads to anxiety. The anxiety that mostly occurs during this period is the anxiety of acquiring a profession or having a career and establishing a good future. In the literature, it is emphasized that university students often experience future and career anxiety during the education period ([12]).

Job selection, plans to take on a role in real life, fear of not finding a job, spouse selection, and various responsibilities can be listed as some of the anxiety-creating factors for students ([36]; [18]). Future events that cause the most anxiety in students can be examined under many headings. Academic and career uncertainty can be expressed as the main reasons for university students’ future anxiety. Students’ fear of failure in their studies or not meeting academic expectations are the primary types of anxiety felt by university students ([78]). In addition, anxiety about finding a job after graduation, choosing the right career, and increasing competition in the job market with recent technological developments, future employment, and career stability can also cause anxiety in students ([3]; [48]). Students’ inability to predict the future due to events such as wars, political conflicts, and international instability contributes to insecurity and negative expectations. Political, economic, and social stresses can increase future anxiety in students ([83]; [3]).

Future anxiety may vary according to students’ individual experiences and socio-cultural context. Negative experiences and failures that students experience throughout their academic lives may cause their future anxiety to increase. Negativities during the university period (exams, friendships, economic problems) may increase the fear of encountering similar situations in the future, which may increase students’ future anxiety. The success criteria determined by students’ families, friends, and environment, and certain expectations of success from them after graduation are the elements that determine future anxiety from a socio-cultural perspective. Students’ social environment and culture affect their future anxiety ([8]; [106]).

### 2.4. Psychological Resilience and Future Anxiety

Psychological resilience plays an important role in coping with stressful and challenging situations and enables adaptation and management of stressful situations ([1]) and a university student with high levels of psychological resilience can better adapt to negative developments and stress factors that arise in school, work, and private life ([22]). They tend to be more optimistic and see everything as a beneficial experience, focusing on personal strengths and qualities ([81]). Thus, their expectations and hopes for the future are equally positive. In this context, it is seen that students’ concerns about the future are an effective factor in their psychological resilience.

Although university students’ negative thinking styles are strong predictors of their anxiety and negative coping methods, there is a negative correlation between positive thinking and anxiety levels ([69]). It is considered that psychological resilience is an important factor in students’ ability to cope with future anxiety, especially during the university period when choices and decisions are made for the future and anxieties arise in parallel. In support of the current literature, [102] ([102]) state that those with high psychological resilience have positive future expectations.

Students with high levels of psychological resilience are more motivated to achieve their goals, produce and implement solutions, have a greater ability to cope with negative emotions and uncertainties and experience less stress ([98]). They adapt more easily and quickly to new situations, and this reduces students’ future anxiety ([2]). In contrast, individuals with low psychological resilience tend to shift to negative emotions in the face of difficulties, often have difficulties responding positively, and as a result, experience future anxiety ([97]). The following hypothesis was developed to determine the relationship between psychological resilience and future anxiety.

**Hypothesis** **1.**
*Psychological resilience has a significant and negative effect on future anxiety.*


### 2.5. Psychological Resilience and Subjective Well-Being

The ability to successfully cope with the encountered difficulties and adapt to new situations healthily, in other words, the ability to adapt to the conditions, are very important in psychological resilience. Adaptation Theory, one of the theories explaining subjective well-being, is also interested in how individuals adapt to new situations. According to the Adaptation Theory, the strong reaction and emotion intensity shown when a new event is encountered for the first time decreases with adaptation to this event over time ([30]). In this context, “Adaptation Theory” shows that there is a relationship between specific well-being and psychological resilience.

It is expected that students with high psychological resilience will be successful in overcoming difficult situations, manage their conscious perceptions well by using positive emotions, and have high subjective well-being. The ability to quickly get rid of negative and difficult situations provides less negative emotions by feeding positive emotions and increases subjective well-being by reducing stress. ([19]). In this context, as students’ psychological resilience levels increase, their subjective well-being levels also increase. It is seen that there is a positive and significant relationship between these two concepts. The following hypothesis was developed to determine the relationship between psychological resilience and subjective well-being.

**Hypothesis** **2:**
*Psychological resilience has a significant and positive effect on subjective well-being.*


### 2.6. Subjective Well-Being and Future Anxiety

Among all subjective well-being theories, the Erek (Telic) Theory also supports the view that there is a relationship between subjective well-being and future anxiety. According to the Erek (Telic) Theory, there are some goals and requirements that individuals have at birth and encounter in the future. As a result of the satisfaction of these needs, happiness occurs, and subjective well-being is achieved. Most of these needs in daily life are met with material goods.

According to the Erek (Telic) Theory, students want to have a good job and build a good future to meet these needs and be happy by reaching satisfaction, in other words, to have subjective well-being. It has been clearly shown in studies that people with high subjective well-being are more likely to earn more money and enjoy their jobs ([27]). Thus, students with high subjective well-being will experience less anxiety about the future. However, the idea that it is necessary to have a job and earn money, and to create a good future by doing a loved job, also leads to future anxiety in students and appears as the biggest concern of university students in the last term of school. Conversely, a person with a low level of subjective well-being is seen to be dissatisfied and unhappy with life, to be depressed, and to have feelings of anger, unease, tension, and anxiety ([29]).

The subjective well-being of students, which is the result of increased positive effects with high life satisfaction, increases their well-being and allows them to have much more control over the future. Thus, their positive feelings increase, they do not worry about the future less, and their stress and fatigue decrease. This situation reveals that students with high subjective well-being experience less anxiety about the future. However, on the contrary, students’ anxiety about the future is negatively related to negative feelings, pessimism, and success ([77]). The extent of this effect depends on factors such as coping strategies, resilience, and social support. The following hypothesis was developed to determine the relationship between subjective well-being and future anxiety.

**Hypothesis** **3.**
*Subjective well-being has a significant and negative effect on future anxiety.*


### 2.7. Mediating Role of Subjective Well-Being

When considering the university period, graduating from school and choosing a profession, all these processes create anxiety about the future in university students. Future anxiety occurs immediately after graduation due to the thought of not being able to find a job for a long time and having difficulty earning a living. It is high during the school period, especially in the last years of university. However, students who are psychologically resilient experience less anxiety about the future and they exhibit a self-confident positivity towards the future ([70]) because these students are less depressed, less stressed by events, and thus less anxious about the future ([51]).

If students have high levels of subjective well-being, which is positively related to future anxiety, they experience positive emotions frequently and negative emotions rarely. They become happier, more competent, and have a more hopeful outlook on the future. Individuals with high subjective well-being are seen to cope better with stress ([26]), and their psychological resilience is also high in parallel with this. When evaluated from this point of view, it can be stated that the existence of subjective well-being as a potential mediator establishes a connection between students’ future anxiety and psychological resilience levels. Based on this approach, the following hypothesis was developed to be investigated and shown in Figure 1.

**Hypothesis** **4.**
*Subjective well-being has a mediating role in the effect of psychological resilience on future anxiety.*


### 2.8. Studies in Turkey

When the research about Türkiye is examined, there is no study in Turkey examining all three concepts related to psychological resilience, future anxiety, and subjective well-being. The studies include examining these concepts in pairs. The first study on psychological resilience in Turkey was conducted by [45] ([45]). As a result of the research, the factors thought to positively affect academic resilience were revealed. Another study conducted by [85] ([85]) studied university students and examined the relationship between psychological resilience and subjective well-being. As a result of the research, it was reported that there was a strong relationship between psychological resilience and subjective well-being. In the study conducted by [90] ([90]), the psychological resilience model regarding subjective well-being was investigated. As a result of this research, it was stated that psychological resilience has an effect on well-being through cognitive evaluation and coping.

In the study conducted by [35] ([35]), the relationship between future anxiety and psychological resilience and life satisfaction of high school students were examined. The study shows that increasing the psychological resilience levels of students can reduce future anxiety. In a study conducted by [10] ([10]) on university students, significant relationships were found between future anxiety and psychological resilience. The research findings showed that students with high levels of psychological resilience had reduced future anxiety.

## 3. Method

### 3.1. Sample and Procedure

The sample of the study consists of students of a state university continuing education in Istanbul. In order to collect data, a meeting was requested with the rector of the relevant university through a mutual friend. In the interview, information was provided about the research and permission was requested for the research to be conducted. Permission was granted by the rector on the condition that the name of the university would not be mentioned, the data would not be shared with third parties, and the data would be collected outside of the lessons. After the necessary permissions were obtained, the data collection process was initiated by the researchers in the social living spaces used by the students. The participants, who were determined by applying the convenience sampling method, were first informed about the research, were assured that anonymity would be protected, were reminded that participation was voluntary, and they could withdraw from the research at any time. The survey took an average of 5 min to complete. During the data collection process, which lasted for two weeks, 501 participants were surveyed face to face. After completing the data collection process, the researchers held a meeting for a preliminary evaluation. In the meeting, it was evaluated that all participants were contacted in Turkish and they were Turkish. Additionally, the surveys were thoroughly examined, and it was decided to exclude 18 surveys with a high amount of missing data due to participant withdrawal from the study. As a result, the data collection process was completed with 483 valid survey responses.

Of the participants, 280 were female and 203 were male; 144 participants were between the ages of 18 and 21, 261 were between the ages of 22 and 25, and 78 were 26 years old and over. In addition, 115 participants were in the first grade, 85 participants were in the second grade, 123 participants were in the third grade, and 160 participants were in the fourth grade.

### 3.2. Statistical Analyses

Analyses were conducted using AMOS 22.0 and SPSS 27.0 software. In the analyses, firstly factor analysis was conducted to control Common Method Bias (CMB) by using SPSS 27.0 software, and then descriptive statistics were conducted to determine the demographic structure. Correlation analysis was applied to determine the relationship between variables. SEM was applied using AMOS 22.0 to determine the fit values of the model. In this approach, many different models are tested to determine the most appropriate model for the data set. Compared to other statistical software, it is frequently used to analyze and verify the suitability of the research model through empirical data, especially in social science research ([13]; [5]; [4]). Process Macro v4.2 (Model 4) was used for hypothesis testing.

### 3.3. Measures

All scales used in the study are unidimensional and self-report scales. These scales, which were developed in the past years and tested for validity/reliability, have also been frequently used in recent studies ([7]; [47]; [72]).

#### 3.3.1. Psychological Resilience Scale

The “Connor–Davidson Resilience Scale (CD-RISC-10)” developed by [16] ([16]) was used to determine the psychological resilience of the participants. The scale, prepared in a 5-point Likert type (“1—not true at all”, “5—true nearly all the time”), has 10 items. Sample items from the scale are “I try to see the humorous side of problems” and “can stay focused under pressure”.

#### 3.3.2. Subjective Well-Being Scale

The Subjective Well-Being (SWB-7) scale developed by [31] ([31]) was used to measure the Subjective Well-Being levels of the participants. The scale, prepared in a 7-point Likert type (1—Strongly Disagree, 7—Strongly Agree), has 5 items. Sample items from the scale are “I am satisfied with my life” and “In most ways my life is close to my ideal”.

#### 3.3.3. Future Anxiety Scale

The Future Anxiety (Dark Future—Short Form) scale developed by [105] ([105]) was used to determine the level of future anxiety in participants. The scale, prepared in a 7-point Likert type (0—Decidedly, 6—Decidedly true), has 5 items. Sample items from the scale are “I am disturbed by the thought that in the future I won’t be able to realize my goals” and “I am afraid that in the future my life will change for the worse”.

#### 3.3.4. Control Variables

Previous studies have shown that psychological variables such as psychological resilience and subjective well-being can vary according to demographic conditions ([72]). In this context, demographic variables (age, education level, and marital status) were determined as control variables in this study, consistent with previous studies.

## 4. Result

### 4.1. Common Method Bias (CMB)

When the data are obtained from a single source and using a survey method, common method bias problems may be encountered. Therefore, several procedural measures were taken to reduce the effects of common method bias in the study. Previously developed scales were used to avoid ambiguity, which is considered as the main source of CBM, and the questions were kept simple and understandable. In order to check the clarity of the statements in the scale, a pre-questionnaire was first conducted on 20 participants and the participants were asked about the clarity of the items in the survey. The feedback received confirmed that the questions were understandable. In addition, participants were reminded not to write their names on the survey and were assured that the data would not be shared with third parties during the survey application. Participation in the survey was voluntary and participants knew that they could withdraw from the study at any time. There was an informative note about the study on the cover page of the survey form. This note included information that there were no right or wrong answers to the items, and there was also information that it was their own ideas that were important. Therefore, it was assumed that they answered the statements in the survey form honestly.

In addition to this precaution, the literature recommends controlling method bias. To determine whether this problem exists, Harman’s single-factor test was conducted in the study. For this, all latent variables were fixed to a single factor and factor analysis was conducted. As a result of the analysis, it was determined that no single factor was dominant and the explained variance values were less than 50% (33.24%). As a result, it was evaluated that there was no problem with common method bias in the study ([44]).

### 4.2. Preliminary Checks

The results of the analyses conducted to determine the validity and reliability of the scales are presented in Table 1.

The results in Table 1 show that the items in the scale have high factor loadings (>0.60) and represent the relevant factor well ([61]; [88]). Factor loadings and CR and AVE values are considered important indicators for the convergent validity of the scales. Accordingly, the fact that AVE values are above 0.50 and CR values are above 0.70 indicates that convergent validity is achieved, and if CR values are greater than AVE values, convergent validity is confirmed ([15]).

In determining discriminant validity, the HTMT criteria proposed by [49] ([49]) and the criteria suggested by [41] ([41]) were used.

According to the criterion suggested by [49] ([49]), HTMT (Heterotrait–Monotrait Ratio) shows the ratios of the average correlations of the expressions belonging to all structures in the study (the heterotrait–heteromethod correlations) to the geometric means of the correlations of the expressions belonging to the same variable (the monotrait–heteromethod correlations). [49] ([49]) stated that the HTMT value should generally be below 0.90 and for concepts that are not close to each other in terms of content should be below 0.85. The results presented in Table 2 show that the HTMT values are within the limit values.

[41] ([41]) state that discriminant validity is achieved when the square root of the AVE value is higher than the correlation of that variable with other variables. As seen in Table 3, this criterion was met for all variables. When all the findings are evaluated together, it is confirmed that the constructs in the study have discriminant validity.

Cronbach’s Alpha (α), McDonald’s Omega (ω), composite reliability (CR), and Average Variance Extracted (AVE) values were checked to determine the reliability of the scales. The literature states that reliability is achieved when Cronbach’s Alpha, McDonald’s Omega, and composite reliability values are greater than 0.7 and the AVE value is greater than 0.5 ([41]). The results in Table 1 confirm that the scales used in the study are reliable.

Finally, the model fit indices were found as χ2/df = 2.042, CFI = 0.94, TLI = 0.95, IFI = 0.96, NFI = 0.95, and RMSEA = 0.049. These index values revealed that the measurement model showed a very good fit.

Before determining the relationship between the variables, a data normality test was performed in parallel with previous studies ([96]) and skewness and kurtosis values were determined for each variable.

The values for skewness and kurtosis being within ±1.5 indicate that the data are normally distributed ([89]). Therefore, the Pearson correlation coefficient was taken into account in the correlation analysis. The analysis results showed that psychological resilience has a positive relationship with subjective well-being (r = 0.487, *p* < 0.05) and a negative relationship with future anxiety (r = −0.329, *p* < 0.05). In addition, the relationship between subjective well-being and future anxiety is also negative (r = −0.547, *p* < 0.05).

### 4.3. Hypothesis Tests

Process Macro v4.2 (Model 4) was used to test the hypotheses. In the simple mediation analysis, psychological resilience was considered as the independent variable (X), subjective well-being as the mediator variable (M), and future anxiety as the outcome variable (Y). The results of the analyses conducted at 95% confidence interval in 5000 resamples are presented in Table 4.

The results in Table 4 show that psychological resilience positively affects subjective well-being (β = 0.326, 95%CI = 0.324, 0.623) and negatively affects future anxiety (β = −0.211; 95%CI = −0.282, −0.109). In addition, subjective well-being negatively affects future anxiety (β = −0.435; 95%CI = −0.421, −0.178). Hypothesis 1, Hypothesis 2, and Hypothesis 3 were supported according to the findings. The final hypothesis of the study is that subjective well-being has a mediating role in the effect of psychological resilience on future anxiety. In order to test this hypothesis, the indirect effect was checked. The findings in Table 4 show that the indirect effect is significant (β = −0.307; 95%CI = −0.324, −0.164). Hypothesis 4 was supported in line with this finding.

## 5. Discussion and Conclusions

This study aims to determine the mediating role of subjective well-being in the relationship between university students’ psychological resilience and future anxiety. University students face many changes and difficulties simultaneously during their education. Students face many negativities, and they are affected by many factors while making important decisions about their future, especially in choosing a profession, choosing a spouse, and achieving their goals as independent individuals. In this case, it is possible to reduce the future anxiety of university students by having high psychological resilience and subjective well-being.

The first hypothesis of the study is “Psychological resilience has a significant and negative effect on future anxiety”. The research proposed and tested that psychological resilience has a significant and negative effect on future anxiety. Our research shows that there is a statistically significant relationship between psychological resilience and future anxiety. In addition, it was observed that students’ psychological resilience has a statistically significant negative effect on future anxiety. This finding shows that students’ future anxiety levels decrease as their psychological resilience increases. The research findings also reveal that students’ psychological resilience provides motivation to cope with negativity and adapt to the current situation after encountering difficulties, and it has a positive effect on future expectations and goals. This naturally reduces future anxiety. This finding is similar to the results of previous studies indicating that psychological resilience negatively affects students’ future anxiety ([1]; [17]; [79]; [35]). Students with high levels of psychological resilience successfully cope with stressful events, communicate well, and easily adapt to their environment when they encounter difficulties ([14]), and in parallel, future anxiety decreases. With these results, it was concluded that the psychological resilience of university students has a negative effect on future anxiety, but it is positive.

In the second hypothesis, it was tested that psychological resilience has a significant and positive effect on subjective well-being. This finding is expected and is a very important result for practitioners. Psychological resilience is positively related to positive emotions, life satisfaction, and productivity. It is negatively related to negative emotions. Studies in the literature have shown that resilience is a strong descriptor of subjective well-being ([34]; [74]; [86]). The result we found in this study that psychological resilience has a significant and positive effect on subjective well-being is parallel to the research findings in the literature ([37]; [52]; [65]; [32]; [60]; [68]; [100]; [107]).

In studies which investigate the relationship between psychological resilience and subjective well-being ([43]; [57]; [92]), it is stated that there is a relationship between positive emotions and psychological resilience, and psychological resilience influences subjective well-being. [39] ([39]) found in their study that psychological resilience and subjective well-being increase together within the scope of the positive emotion relationship. These findings are consistent with the hypothesis that we put forward in our study.

Psychological resilience is achieving positive and unexpected results in the face of negative situations and adapting to extraordinary conditions. Individuals’ subjective well-being levels are also shaped according to this adaptation process. The basic structure of the Adaptation Theory is formed by individuals’ adaptation or adjustment to positive or negative living conditions. If events are better than the individual’s current situation, the individual will be happy; if this situation continues, the individual will become used to it and will create a new criterion to evaluate events. In this respect, the Adaptation Theory supports the research findings that psychological resilience has a significant and positive effect on subjective well-being. Psychological resilience is positively related to positive emotions and negatively related to negative emotions. Students with high psychological resilience are better able to overcome the negative situations they encounter throughout their education life and adapt better to the new situation.

The third hypothesis examined whether subjective well-being has a significant and negative effect on students’ future anxiety. It was found that there was a negative relationship between students’ future anxiety and subjective well-being levels. Students with high subjective well-being have more positive emotions and look to the future with more hope. It is known that individuals with low future anxiety levels have high life satisfaction, psychological resilience, and subjective well-being levels ([33]). This result we found in our study is similar to other studies ([66]; [38]; [104]; [24]). Having a positive perspective on the future is important in terms of providing insight into how they feel about their own lives and future ([58]). Thus, it will affect them to have more positive expectations in their future planning and shed light on their goals. In addition, it seems that when evaluating attitudes towards the future, they should be evaluated together with subjective well-being.

According to the Erek (Telic) Theory, the source of subjective well-being is the satisfaction of individuals’ desires and needs and their ability to achieve their goals. At this point, students who have achievable individual goals and act in line with these goals in a way that will make their daily lives easier can increase their well-being. When the most important goals for students are considered to be graduating from school, finding a job, having a good career, and becoming financially independent, achieving these goals will increase their well-being and thus reduce their anxiety about the future. Erek (Telic) Theory supports research findings that show a negative relationship between students’ anxiety about the future and their subjective well-being levels. This is because students with high subjective well-being have more positive emotions and are more hopeful about the future.

The fourth hypothesis of our study examined the mediating role of subjective well-being in the effect of psychological resilience on future anxiety. Our study is the first study in Türkiye to address subjective well-being in the relationship between psychological resilience and future anxiety. This is the first attempt to investigate the mediating role of the cognitive triad between psychological resilience and future anxiety. According to the results obtained in the study, it was found that subjective well-being has a mediating role in the relationship between psychological resilience and future anxiety. When students’ subjective well-being is high, their happiness increases, and their positive emotions increase compared to negative emotions. This will prevent them from worrying about the future and reduce their future anxiety. It is seen that students with high subjective well-being have less future anxiety and more positive emotions, and also have high psychological resilience because psychologically healthy students overcome the difficulties they encounter and adapt better to new situations. This finding of our research is also consistent with a limited number of studies examining the mediating role of subjective well-being ([20]; [28]; [53]; [26]). In fact, this finding shows to what extent students’ subjective well-being is an important variable for the relationship between psychological resilience and future anxiety, and this is supported by the literature and this study. Therefore, subjective well-being should be considered important along with psychological resilience, which are among the factors that will reduce future anxiety in university students, and steps should be taken to improve them.

Students with high levels of psychological resilience cope more successfully with stressful academic events. This affects their emotional state positively, thus it causes their subjective well-being to be high. They also work harder in their academic lives. This will make them more likely to have positive emotions and happiness in their education, causing them to experience less anxiety about the future.

However, difficulties and methodological limitations that arise during the conduct of this research may affect theoretical results. The fact that the sample in this study is limited to only one university student may cause a representativeness problem. Thus, it may lead to questioning whether the theoretical contributions presented by the research are valid for different samples or contexts ([56]). Factors such as social desirability bias in the surveys used in this study may cause the contributions predicted by the theoretical model not to occur in real life, not to reflect fully, or to see different results than expected.

## 6. Managerial Implications

When the results of this study are evaluated in general, the psychological resilience of university students directly affects their subjective well-being and future anxiety. It also indirectly affects future anxiety through subjective well-being. For this reason, it can be concluded that the subjective well-being of university students who have psychological resilience will also be high. It can be said that high subjective well-being will make it easy for students to overcome the negative or difficult situations they will encounter by taking the necessary steps to cope with them. This will also prevent their future anxiety from increasing when they face difficult and negative situations. In addition, it can be expressed that students with low future anxiety have high subjective well-being. As a result, it positively affects their psychological resilience levels or it is a protective factor on their psychological resilience.

## 7. Limitations and Future Research

This study is the first study in Turkiye to examine the relationship between psychological resilience and future anxiety in the context of subjective well-being. While it is considered to contribute to the field in this regard, it also has certain limitations. The sample consists of a limited number of participants and is restricted to students from only one state university. So the generalizability of the results is limited. Therefore it is suggested that future research could be conducted with larger and more diverse sample groups, and other types of universities, cities, or cultures for greater generalizability of the results. In addition, the results obtained from this study are limited to the participants’ own responses.

Since the research was conducted with university students, the generalizability of the results is limited. The fact that the sample consists solely of university students may lead to reduced generalizability of the findings. Thus, the obtained results may not be applicable to the entire population ([56]). Additionally, the use of the convenience sampling method and the collection of data through self-reported surveys may result in participants providing dishonest responses, which could distort the findings ([76]). Despite the above limitations, the findings of the current study are encouraging.

This study has several important implications for policy initiatives, professional practices, and research efforts within the scope of academic research on psychological resilience. For students to best manage the negativities they or their relatives encounter, they must first be carefully managed and guided. Governments should provide community-based opportunities which provide students with access to both environmental and personal resources to improve their resilience in meaningful ways ([35]; [94]). Within the scope of programs for universities, psychologists should work to strengthen protective and supportive factors that will help students proactively develop psychological resilience. The importance of social support that will help university students to reduce their future anxiety should be explained, and psychological and physical studies should be carried out to support both subjective well-being and psychological resilience. In order to determine the main reasons that negatively affect students’ psychological resilience, in-depth interviews should be conducted with students and special solutions should be developed based on the reasons identified.

As examples of policy-related initiatives, educational campaigns and mentoring programs can be organized for students. Making future and career centers more active and activities more widespread, especially for senior students, can be considered among the activities that can reduce students’ future anxiety. In this way, students who try to achieve future goals and experience a certain amount of anxiety in the process will be more likely to make decisions with less anxiety and higher confidence about their future ([54]; [2]). According to these results, psychosocial intervention programs can be prepared within the scope of psychological counseling and guidance practices in universities to reduce students’ future concerns. The findings obtained in the research can also be used for psychological counseling and guidance services that contribute positively to the subjective well-being of individuals. In addition, education politicians and university administrations can establish counseling centers within the university where students can share all their problems and increase their psychological resilience. Training, seminars, and events aimed at increasing psychological resilience can be provided. Students can be supported for future issues with activities such as career days and job opportunities in the last semesters of the university.

In this study, the mediating role of subjective well-being was also examined. In future studies, the mediating or moderating role of other variables that may affect the relationship between psychological resilience and future anxiety can be examined as well. In addition, advanced statistics can be conducted to determine the latent and mediating variables that may be effective.

## Figures and Tables

**Figure 1 behavsci-15-00244-f001:**
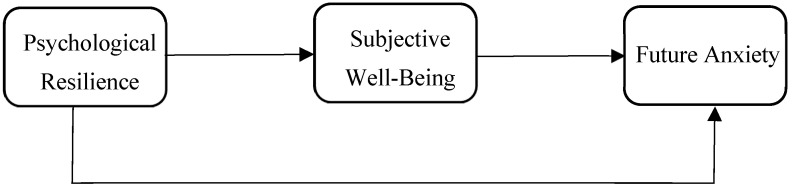
Research model.

**Table 1 behavsci-15-00244-t001:** Validity and reliability results.

Expressions	Factor Loading	Cronbach’s Alpha	McDonald’s ω	CR	AVE
Psychological Resilience		0.912	0.910	0.908	0.501
PR1	0.683				
PR2	0.725				
PR3	0.689				
PR4	0.740				
PR5	0.759				
PR6	0.602				
PR7	0.771				
PR8	0.785				
PR9	0.692				
PR10	0.608				
Subjective Well-Being		0.836	0.833	0.833	0.503
SWB1	0.633				
SWB2	0.659				
SWB3	0.788				
SWB4	0.644				
SWB5	0.802				
Future Anxiety		0.882	0.881	0.880	0.514
DF1	0.746				
DF2	0.820				
DF3	0.677				
DF4	0.609				
DF5	0.713				
DF6	0.799				
DF7	0.632				
Model Fit Values			CMIN/DF = 2.042; CFI = 0.94; TLI = 0.95; IFI = 0.96; RMSEA = 0.049		

**Table 2 behavsci-15-00244-t002:** Discriminant validity results (HTMT criterion).

Variables	Psychological Resilience	Subjective Well-Being	Future Anxiety
Psychological Resilience	^-^		
Subjective Well-Being	0.536	-	
Future Anxiety	0.482	0.623	-

**Table 3 behavsci-15-00244-t003:** Discriminant validity results (Fornell and Larcker criterion).

Variables	Mean	SD	1	2	3
Psychological Resilience	3.37	0.86	0.708 ^a^		
Subjective Well-Being	3.49	1.04	0.487 **	0.709 ^a^	
Future Anxiety	3.02	1.45	−0.329 **	−0.547 **	0.717 ^a^
Skewness			−1.251	0.657	−1.152
Courtois			0.520	−0.381	1.058

** *p* < 0.05, ^a^ is the square root of AVE.

**Table 4 behavsci-15-00244-t004:** Hypothesis test results.

Paths	ß	S.H	LLCI	ULCI
Psychological Resilience → Subjective Well-Being	0.326	0.086	0.324	0.623
Psychological Resilience → Future Anxiety	−0.211	0.062	−0.282	−0.109
Subjective Well-Being → Future Anxiety	−0.435	0.058	−0.421	−0.178
Indirect Effect				
PR → SWB → DF	−0.307	0.349	−0.324	−0.164

PR: Psychological Resilience, SWB: Subjective Well-Being.

## Data Availability

The data of this study are available upon reasonable request from the corresponding author.

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
