# Peer review of "Psychological Resilience and Future Anxiety Among University Students: The Mediating Role of Subjective Well-Being"

_behavsci, 2025, doi:10.3390/bs15030244_

Round 1

Reviewer 1 Report

Comments and Suggestions for Authors

1. Please start the abstract with one or two sentences about the study's background and importance to set the stage for readers. The term 'dark future' may not be familiar in academic contexts and could use clarification or an alternative. Focus the abstract on the study's contributions and implications rather than detailed methodology, which is currently missing.

2. The paper lacks a literature review section, which is essential for setting the context and framing the narrative of the research. Particularly, it would be beneficial to include a discussion on relevant events or studies in Turkey, as they seem pivotal to the study’s context.

3. The theoretical foundation of the study needs strengthening. A detailed theoretical model, supporting the research hypotheses, appears to be missing and is crucial for establishing a robust framework.

4. The choice of using AMOS for SEM analysis requires justification, particularly why it was preferred over other statistical software. Additionally, an introduction to SEM and CB-SEM is necessary for clarity. The methodology section also lacks a detailed description of the data collection processes and procedures.

5. findings: According to “Factors Influencing Students ’ Continuous Intentions for Using Micro-Lectures in the Post-COVID-19 Period : A Modification of the UTAUT-2 Approach,” Electron., vol. 11, no. 1924, 2022. Author may add Normality testing and descriptive statistics.

6. please provide HTMT and fornel-lacker for discriminant validity.

7. The implications section fails to outline the theoretical contributions of the study. It would be beneficial if the author could explicitly address this aspect, perhaps explaining any limitations or challenges encountered that might have affected the theoretical outcomes.

8. The current reference list shows an excessive number of citations to works by Sürücü, which could raise concerns about citation bias. Diversifying the sources cited could enhance the credibility of the paper.

Author Response

Comments 1:  Please start the abstract with one or two sentences about the study's background and importance to set the stage for readers. The term 'dark future' may not be familiar in academic contexts and could use clarification or an alternative. Focus the abstract on the study's contributions and implications rather than detailed methodology, which is currently missing.

Response 1: Thank you pointing out this point. We agree with this comment. Therefore as you said, we start abstract with information about the study's background and importance to set the stage for readers. Also we changed term “dark future” with the term “future anxiety” in the whole context since 'dark future' may not be familiar in academic contexts.  In addition, we deleted detailed methodology to explain contributions and implications in abstract. This change can be found page 1, paragraph 1, and line 12.

Future Anxiety is the worry and concern individuals experience regarding uncertainties and potential negative outcomes in their future. This emotional state can manifest at different stages of students' academic lives and can impact their academic performance and social relationships. In the process of coping with negative experiences and overcoming challenges, psychological resilience plays a crucial role. Students who struggle to manage stress and have high levels of anxiety tend to experience future anxiety more intensely. The aim of this study is to determine the mediating role of subjective well-being in the relationship between psychological resilience and future anxiety among university students. The study was conducted with a total of 483 university students, including 280 females and 203 males. Data were collected using the Connor-Davidson Resilience Scale (CD-RISC-10), Subjective Well-Being Scale (SWB-7), and Future Anxiety (Dark Future) Scale (Short Form). Analyses were performed using AMOS 22.0 and SPSS 27.0 software. The findings indicate that psychological resilience has a significant negative effect on future anxiety, a significant positive effect on subjective well-being, and that subjective well-being has a significant negative effect on future anxiety. Additionally, the study found that subjective well-being mediates the relationship between psychological resilience and future anxiety.

Comments 2:  The paper lacks a literature review section, which is essential for setting the context and framing the narrative of the research. Particularly, it would be beneficial to include a discussion on relevant events or studies in Turkey, as they seem pivotal to the study’s context.

Response 2: Thank you pointing out this point. We agree with this comment. First of all, we tried to give literature review to understand about our subject, but we did not mention about studies. We could mention about studies and therefore we included studies in Turkey. This change can be found page 7, and line 336.

When the researches about Türkiye are examined, there is no study in Turkey examining all three concepts related to psychological resilience, future anxiety and subjective well-being. The studies include examining these concepts in pairs. The first study on psychological resilience in Turkey was conducted by Gizir (2004). As a result of the research, the factors thought to positively affect academic resilience were revealed. Another study conducted by SiviÅŸ-Çetinkaya (2013) studied university students and examined the relationship between psychological resilience and subjective well-being. As a result of the research, it was reported that there was a strong relationship between psychological resilience and subjective well-being. In the study conducted by Terzi (2005), the psychological resilience model regarding subjective well-being was investigated. As a result of this research, it was stated that psychological resilience has an effect on well-being through cognitive evaluation and coping.

In the study conducted by Dursun and Özkan (2019), the relationship between future anxiety and psychological resilience and life satisfaction of high school students were examined. The study shows that increasing the psychological resilience levels of students can reduce future anxiety. In a study conducted by Biltekin and Duman (2024) on university students, significant relationships were found between future anxiety and psychological resilience. The research findings showed that students with high levels of psychological resilience had reduced future anxiety.

Comments 3:  The theoretical foundation of the study needs strengthening. A detailed theoretical model, supporting the research hypotheses, appears to be missing and is crucial for establishing a robust framework.

Response 3: Thank you pointing out this point. We agree with this comment. We gave more detailed information about theoretical model to support the research hypotheses. 3 paragraphs that we added In addition to existing hypotheses are presented below. This change can be found page 5 and 6, line 245, 266 and 296.

Psychological Resilience ve Future Anxiety

Students with high levels of psychological resilience are more motivated to achieve their goals, produce and implement solutions, have greater ability to cope with negative emotions and uncertainties, and experience less stress (Xu et al., 2020), adapt more easily and quickly to new situations, and this reduces students' future anxiety (Aksu & Kuas, 2024). In contrast, individuals with low psychological resilience tend to shift to negative emotions in the face of difficulties, often have difficulty responding positively, and as a result, experience future anxiety (Xiang et al., 2022). The following hypothesis was developed to determine the relationship between psychological resilience and future anxiety.

Psychological Resilience ve Subjective Well-Being

It is expected that students with high psychological resilience will be successful in overcoming difficult situations, manage their conscious perceptions well by using positive emotions, and have high subjective well-being. Because the ability to quickly get rid of negative and difficult situations provides less negative emotions by feeding positive emotions, and increases subjective well-being by reducing stress. (Çelebi, 2023). In this context, as students' psychological resilience levels increase, their subjective well-being levels also increase. It is seen that there is a positive and significant relationship between these two concepts. The following hypothesis was developed to determine the relationship between psychological resilience and subjective well-being.

Subjective Well-Being ve Future Anxiety

The subjective well-being of students, which is the result of increased positive effects with high life satisfaction, increases their well-being and allows them to have much more control over the future. Thus, their positive feelings increase, they do not worry about the future less, and their stress and fatigue decrease. This situation reveals that students with high subjective well-being experience less anxiety about the future. However, on the contrary, students' anxiety about the future is negatively related to negative feelings, pessimism and success (Öztekin, 2025). The extent of this effect depends on factors such as coping strategies, resilience and social support. The following hypothesis was developed to determine the relationship between subjective well-being ve future anxiety.

Comments 4:  The choice of using AMOS for SEM analysis requires justification, particularly why it was preferred over other statistical software. Additionally, an introduction to SEM and CB-SEM is necessary for clarity. The methodology section also lacks a detailed description of the data collection processes and procedures.

Response 4: Thank you pointing out this point. We agree with this comment. We gave information about a detailed description of the data collection processes and procedures, the choice of using AMOS for SEM analysis and an introduction to SEM and CB-SEM and. We added this information to that section. This change can be found page 7, line 357 and 383.

The sample of the study consists of students of a state university continuing educa-tion in Istanbul. In order to collect data, a meeting was requested with the rector of the rel-evant university through a mutual friend. In the interview, information was provided about the research and permission was requested for the research to be conducted. Per-mission was granted by the rector on the condition that the name of the university would not be mentioned, the data would not be shared with third parties and the data would be collected outside of the lessons. After the necessary permissions were obtained, the data collection process was initiated by the researchers in the social living spaces used by the students. The participants, who were determined by applying the convenience sampling method, were first informed about the research, assured that anonymity would be pro-tected, reminded that participation was voluntary and they could withdraw from the re-search at any time. The survey took an average of 5 minutes to complete. During the data collection process, which lasted for two weeks, 501 participants were surveyed face to face. After completing the data collection process, the researchers held a meeting for a prelimi-nary evaluation. In the meeting, it was evaluated that all participants were contacted in Turkish and they were Turkish. Additionally, the surveys were thoroughly examined, and it was decided to exclude 18 surveys with a high amount of missing data due to partici-pant withdrawal from the study. As a result, the data collection process was completed with 483 valid survey responses.

Analyses were conducted using AMOS 22.0 and SPSS 27.0 software. In the analyses, firstly factor analysis was conducted to control Common Method Bias (CMB) by uisng  SPSS 27.0 software, then descriptive statistics were conducted to determine the demographic structure. Correlation analysis was applied to determine the relationship between variables. SEM was applied using AMOS 22.0 to determine the fit values ​​of the model. In this approach, many different models are tested to determine the most appropriate model for the data set. Compared to other statistical software, it is frequently used to analyze and verify the suitability of the research model through empirical data, especially in social science research (Brown, 2015; Anthony et al., 2021; Almulla & Alamri, 2021). Process Macro v4.2 (Model 4) was used for hypothesis testing.

Comments 5:  findings: According to “Factors Influencing Students ’ Continuous Intentions for Using Micro-Lectures in the Post-COVID-19 Period : A Modification of the UTAUT-2 Approach,” Electron., vol. 11, no. 1924, 2022. Author may add Normality testing and descriptive statistics.

Response 5: Thank you pointing out this point. We agree with this comment. We added Normality testing and descriptive statistics and gave information about it. This change can be found page 11, line 490 and 497.

Before determining the relationship between the variables, data normality test was performed in parallel with previous studies (Wijaya & Weinhandl, 2022) and skewness and courtois values ​​were determined for each variable.

The values ​​for skewness and kurtosis being within ±1.5 indicate that the data are normally distributed (Tabachnick & Fidell, 2013). Therefore, the Pearson correlation coefficient was taken into account in the correlation analysis.

Comments 6:  . please provide HTMT and fornel-lacker for discriminant validity.

Response 6: Thank you pointing out this point. We agree with this comment. We provide information about HTMT and fornel-lacker for discriminant validity below. And other information were added to text. This change can be found page 7, line 463.

In determining discriminant validity, the HTMT criteria proposed by Henseler et al. (2015) and the criteria suggested by Fornell and Larcker (1981) were used.

According to the criterion suggested by Henseler et al. (2015), HTMT (HeterotraitMonotrait Ratio) shows the ratios of the average correlations of the expressions belonging to all structures in the study (the heterotrait-heteromethod correlations) to the geometric means of the correlations of the expressions belonging to the same variable (the monotrait-heteromethod correlations). Henseler et al. (2015) stated that the HTMT value should generally be below 0.90 and for concepts that are not close to each other in terms of content should be below 0.85. The results presented in Table 2 show that the HTMT values ​​are within the limit values.

Variables

Psychological Resilience

Subjective Well-Being

Future

Anxiety

Psychological Resilience

-

Subjective Well-Being

0.536

-

Future Anxiety

0.482

0.623

-

Comments 7:  The implications section fails to outline the theoretical contributions of the study. It would be beneficial if the author could explicitly address this aspect, perhaps explaining any limitations or challenges encountered that might have affected the theoretical outcomes.

Response 7: Thank you pointing out this point. We agree with this comment. We addressed this aspects explaining any limitations or challenges encountered that might have affected the theoretical outcomes. We added this paragraphes. This change can be found page 14, line 623.

Students with high levels of psychological resilience cope more successfully with stressful academic events. This affects their emotional state positively, thus causing their subjective well-being to be high. Students with high subjective well-being work harder in their academic lives. This will make them more likely to have positive emotions and happiness in their education, causing them to experience less anxiety about the future.

However, difficulties and methodological limitations that arise during the conduct of the research may affect theoretical results. The fact that the sample in this study is limited to only one university student may cause a representativeness problem. Thus, it may lead to questioning whether the theoretical contributions presented by the research are valid for different samples or contexts (Karaçam, 2015). Factors such as social desirability bias in the surveys used in this study may cause the contributions predicted by the theoretical model not to occur in real life, not to reflect fully, or to see different results than expected.

Comments 8:  The current reference list shows an excessive number of citations to works by Sürücü, which could raise concerns about citation bias. Diversifying the sources cited could enhance the credibility of the paper.

Response 8: Thank you pointing out this point. We agree with this comment. We Diversified the sources cited could enhance the credibility of the paper and deleted 2 citations to works by Sürücü.

Please see the attachment. I uploaded an attachment whole revised text with red marked.

Reviewer 2 Report

Comments and Suggestions for Authors

The study is original and valuable, as it contributes new evidence to existing research. Therefore, we congratulate the authors. However, we believe there are aspects that could help improve the article. These include:

  • Deepening the discussion on practical implications: While the study mentions the practical implications for student mental health and educational policies, further exploration is needed on how these implications can be translated into concrete actions.

  • Considering sample diversity: Although the study's sample is substantial (483 students), it is limited to students from a state university in Istanbul. It could be suggested that future research include more diverse samples, considering other types of universities, geographic regions, or cultures. This would allow for greater generalizability of the results.

  • Exploring other moderating or mediating variables: The study focuses on the mediating role of subjective well-being. It would be interesting to suggest that future research explore other variables that could moderate or mediate the relationship between psychological resilience and a pessimistic view of the future.

  • Providing more context on the sample: While it is mentioned that the participants are university students from a state university in Istanbul, it could be suggested that more information be provided about the specific characteristics of the sample, such as the types of degrees they are pursuing, their socioeconomic status, or ethnic background. This could help better interpret the results and their applicability.

  • Addressing the complex nature of the "dark future": The study defines "dark future" as a pessimistic view of the future. However, it could be suggested that the different dimensions and nuances that may compose this view be explored in greater depth, especially as it relates to anxiety and personal and global fears. What types of future events generate the most anxiety in students? How does this anxiety vary based on their personal experiences and context?

  • Strengthening the theoretical discussion: Although the study relies on theories such as the Adaptation Theory and Erek (Telic) Theory, it could be suggested that a deeper discussion is provided on how these theories relate to the study’s findings. This could enrich the theoretical framework and offer a more sophisticated understanding of the phenomena under investigation.

  • Expanding the discussion on limitations: Although the study acknowledges some limitations, such as the generalizability of the results, it could be suggested that the discussion be broadened to include possible methodological limitations and how they may have affected the results.

  • Considering the influence of external factors: The study acknowledges that students are affected by external factors such as wars, natural disasters, and economic crises. It could be suggested that the exploration of how these factors influence resilience and subjective well-being, and consequently affect a pessimistic view of the future, be further explored.

Comments on the Quality of English Language

Although the English is generally good, some phrases could benefit from a review by a native speaker or an expert in academic writing to ensure optimal fluency and naturalness.

In some instances, the use of certain connectors or prepositions could be revised for greater precision.

Some sentences may become somewhat lengthy and could be divided to improve clarity.

Author Response

Comments 1:  Deepening the discussion on practical implications: While the study mentions the practical implications for student mental health and educational policies, further exploration is needed on how these implications can be translated into concrete actions.

Response 1: Thank you pointing out this point. We agree with this comment. Therefore we give additional explanations below  how these implications can be translated into concrete actions. We have included this explanation in that paragraph. This change can be found page 15, line 672.

Within the scope of programs for universities, psychologists should work to strengthen protective and supportive factors that will help students proactively develop psychological resilience. The importance of social support that will help university students reduce their future anxiety should be explained, and psychological and physical studies should be carried out to support both subjective well-being and psychological resilience. In order to determine the main reasons that negatively affect students' psychological resilience, in-depth interviews should be conducted with students and special solutions should be developed based on the reasons identified.

Comments 2:  Considering sample diversity: Although the study's sample is substantial (483 students), it is limited to students from a state university in Istanbul. It could be suggested that future research include more diverse samples, considering other types of universities, geographic regions, or cultures. This would allow for greater generalizability of the results.

Response 2: Thank you pointing out this point. We agree with this comment. Therefore, as you said, we have included additional suggestions for future research and revised this paragraph with mentioned suggestions. This change can be found page 14, line 649.

The study is the first study in Turkiye to examine the relationship between psychological resilience and future anxiety in the context of subjective well-being. While it is considered to contribute to the field in this regard, it also has certain limitations. The sample consists of a limited number of participants and is restricted to students from only one state university. So the generalizability of the results is limited. Therefore it is suggested that future research could be conducted with larger and more diverse sample groups, other types of universities, cities or cultures for greater generalizability of the results. In addition, the results obtained from this study are limited to the participants' own re-sponses.

Comments 3:  Exploring other moderating or mediating variables: The study focuses on the mediating role of subjective well-being. It would be interesting to suggest that future research explore other variables that could moderate or mediate the relationship between psychological resilience and a pessimistic view of the future.

Response 3: Thank you pointing out this point. We agree with this comment. Therefore, as you said, we have mentioned and included about other variables that could moderate or mediate the relationship between psychological resilience and future anxiety for future researches. This change can be found page 15, line 696.

In this study, the mediating role of subjective well-being was examined. In future studies, the mediating or moderating role of other variables that may affect relationship between psychological resilience and future anxiety can also be examined. In addition, advanced statistics can be conducted to determine the latent and mediating variables that may be effective.

Comments 4:  Providing more context on the sample: While it is mentioned that the participants are university students from a state university in Istanbul, it could be suggested that more information be provided about the specific characteristics of the sample, such as the types of degrees they are pursuing, their socioeconomic status, or ethnic background. This could help better interpret the results and their applicability.

Response 4: Thank you pointing out this point. We agree with this comment. We had information also about their ethnic background. This change can be found page 7, line 370.

After completing the data collection process, the researchers held a meeting for a preliminary evaluation. In the meeting, it was evaluated that all participants were contacted in Turkish and they were Turkish.

Comments 5:  Addressing the complex nature of the "dark future": The study defines "dark future" as a pessimistic view of the future. However, it could be suggested that the different dimensions and nuances that may compose this view be explored in greater depth, especially as it relates to anxiety and personal and global fears. What types of future events generate the most anxiety in students? How does this anxiety vary based on their personal experiences and context?

Response 5: Thank you pointing out this point. We agree with this comment. Therefore we give informations about future anxiety and fears below.  We have included this explanation. This change can be found page 5, line 204.

Job selection, plans to take on a role in real life, fear of not finding a job, spouse selection and various responsibilities can be listed as some of the anxiety-creating factors for students (Dursun & Aytaç, 2009; Çakmak & Hevedanlı, 2005). Future events that cause the most anxiety in students can be examined under many headings. Academic and career uncertainty can be expressed as the main reasons for university students' future anxiety. Students' fear of failure in their studies or not meeting academic expectations are the primary types of anxiety felt by university students (Öztekin et al., 2025). In addition, anxiety about finding a job after graduation, choosing the right career and increasing competition in the job market with recent technological developments, future employment and career stability can also cause anxiety in students (Al-Baddai et al., 2021; Hammad, 2016). Students' inability to predict the future due to events such as wars, political conflicts and international instability contributes to insecurity and negative expectations. Political, economic and social stresses can increase future anxiety in students (Sart, 2023; Al-Baddai et al., 2021).

Future anxiety may vary according to students’ individual experiences and socio-cultural context. Negative experiences and failures that students experience throughout their academic lives may cause their future anxiety to increase. Negativities during the university period (exams, friendships, economic problems) may increase the fear of encountering similar situations in the future, which may increase students’ future anxiety. The success criteria determined by students’ families, friends and environment, and certain expectations of success from them after graduation are the elements that determine future anxiety from a socio-cultural perspective. Students’ social environment and culture affect their future anxiety (Aygün, 2014; Zengin & Åžengel, 2020).

Comments 6:  Strengthening the theoretical discussion: Although the study relies on theories such as the Adaptation Theory and Erek (Telic) Theory, it could be suggested that a deeper discussion is provided on how these theories relate to the study’s findings. This could enrich the theoretical framework and offer a more sophisticated understanding of the phenomena under investigation.

Response 6: Thank you pointing out this point. We agree with this comment. Therefore we give informations about how these theories relate to the study’s findings. below.  We have included this explanation. This change can be found page 13, line 568 and 592.

Adaptation Theory

Psychological resilience is achieving positive and unexpected results in the face of negative situations and adapting to extraordinary conditions. Individuals' subjective well-being levels are also shaped according to this adaptation process. The basic structure of the Adaptation Theory is formed by individuals' adaptation or adjustment to positive or negative living conditions. If events are better than the individual's current situation, the individual will be happy; if this situation continues, the individual will get used to it and will create a new criterion to evaluate events. The Adaptation Theory also supports the research findings that psychological resilience has a significant and positive effect on subjective well-being. Because psychological resilience is positively related to positive emotions and negatively related to negative emotions. Students with high psychological resilience are better able to overcome the negative situations they encounter throughout their education life and adapt better to the new situation.

Erek (Telic) Theory

According to the Erek (Telic) Theory, the source of subjective well-being is the satisfaction of individuals’ desires and needs and their ability to achieve their goals. At this point, students having achievable individual goals and acting in line with these goals in a way that will make their daily lives easier can increase their well-being. When the most important goals for students are considered to be graduating from school, finding a job, having a good career and becoming financially independent, achieving these goals will increase their well-being and thus reduce their anxiety about the future. Erek (Telic) Theory supports research findings that show a negative relationship between students’ anxiety about the future and their subjective well-being levels. This is because students with high subjective well-being have more positive emotions and are more hopeful about the future.

Comments 7:  Expanding the discussion on limitations: Although the study acknowledges some limitations, such as the generalizability of the results, it could be suggested that the discussion be broadened to include possible methodological limitations and how they may have affected the results.

Response 7: Thank you pointing out this point. We agree with this comment. Therefore we give informations about discussion and broadened including possible methodological limitations and how they may have affected the results. This change can be found page 14, line 658.

Since the research was conducted with university students, the generalizability of the results is limited. The fact that the sample consists solely of university students may lead to reduced generalizability of the findings. Thus, the obtained results may not be applicable to the entire population (Karaçam, 2015). Additionally, the use of the convenience sampling method and the collection of data through self-reported surveys may result in participants providing dishonest responses, which could distort the findings (Özkan & Kaya, 2015).

Comments 8:  Considering the influence of external factors: The study acknowledges that students are affected by external factors such as wars, natural disasters, and economic crises. It could be suggested that the exploration of how these factors influence resilience and subjective well-being, and consequently affect a pessimistic view of the future, be further explored.

Response 8: Thank you pointing out this point. We agree with this comment. Therefore we give informations about how these factors influence resilience and subjective well-being, and consequently affect a pessimistic view of the future. This change can be found page 3, line 105.

In general, psychological resilience of students includes showing flexibility in the face of negative events, staying strong and adapting to new situations. However, achieving and maintaining this situation is affected by many factors. These factors are external factors that express environmental and social effects such as wars, natural disasters, economic crises (Fletcher & Sarkar, 2013; Masten et al., 2015) as mentioned above. The environment of uncertainty created by these environmental factors causes students’ psychological resilience and subjective well-being to be low. Students’ concerns about their future increase due to the effect of these negative developments and changes. Therefore, it is important a stable and safe environment for students. The fact that students have a stable and safe environment during school years contributes to the development of their resilience and, consequently, their subjective well-being. This situation is associated with a high level of psychological resilience and psychological well-being (Davydov et al., 2010).

Comments 9:  Comments on the Quality of English Language

Although the English is generally good, some phrases could benefit from a review by a native speaker or an expert in academic writing to ensure optimal fluency and naturalness. In some instances, the use of certain connectors or prepositions could be revised for greater precision. may become somewhat lengthy and could be improve clarity.

Response 9: Thank you pointing out this point. We agree with this comment. Therefore we used some phrases to ensure optimal fluency and naturalness. We divided some sentences and checked text to again. Final version of text was added.

Please see the attachment. I uploaded an attachment whole revised text with red marked.

Round 2

Reviewer 1 Report

Comments and Suggestions for Authors

dear editor, dear author,

I have read the revised manuscript and found that the author has made comprehensive improvements across all sections, from the title to the references. I have not found any weaknesses; the manuscript is now ready for publication.

Well done.